# Health Research and Education during and after the COVID-19 Pandemic: An Australian Clinician and Researcher Perspective

**DOI:** 10.3390/diagnostics13020289

**Published:** 2023-01-12

**Authors:** Dennis J. Cordato, Kaneez Fatima Shad, Wissam Soubra, Roy G. Beran

**Affiliations:** 1Department of Neurophysiology, Liverpool Hospital, Locked Bag 7103, Liverpool BC, Sydney, NSW 1871, Australia; 2Ingham Institute for Applied Medical Research, Campbell St Liverpool, Sydney, NSW 2170, Australia; 3South Western Sydney Clinical School, University of NSW, Liverpool, Sydney, NSW 2170, Australia; 4School of Life Sciences, University of Technology, Ultimo, Sydney, NSW 2007, Australia; 5Faculty of Health Sciences, Australian Catholic University, 40 Edward St, Sydney, NSW 2060, Australia; 6School of Medical Sciences, ISRA, University of Hyderabad Pakistan, Hyderabad 71000, Pakistan; 7A Healthy Step Podiatry, Lakemba, Sydney, NSW 2195, Australia; 8School of Medicine, Griffith University, Southport, QLD 4222, Australia; 9School of Medicine, Western Sydney University, Sydney, NSW 2170, Australia; 10Faculty of Sociology, Sechenov Moscow First State University, Moscow 119991, Russia

**Keywords:** education, research, adaptation, methodology, trials, telehealth, COVID-19

## Abstract

**Introduction:** The COVID-19 pandemic had an unprecedented global effect on teaching and education. This review discusses research, education and diagnostics from the perspectives of four academic clinicians and researchers across different facilities in Australia. **Materials and methods:** The study adopted a literature review and an Australian researcher’s perspective on the impact of the COVID-19 pandemic on health education, research and diagnostics. **Results:** At the start of the pandemic, medical facilities had to adhere urgently to major work restrictions, including social distancing, mask-wearing rules and/or the closure of facilities to protect staff, students and patients from the risk of COVID-19 infection. Telemedicine and telehealth services were rapidly implemented and adapted to meet the needs of medical education, the teaching of students, trainee doctors, nursing and allied health staff and became a widely accepted norm. The impact on clinical research and education saw the closure of clinical trials and the implementation of new methods in the conducting of trials, including electronic consents, remote patient assessments and the ability to commence fully virtual clinical trials. Academic teaching adapted augmented reality and competency-based teaching to become important new modes of education delivery. Diagnostic services also required new policies and procedures to ensure the safety of personnel. **Conclusions:** As a by-product of the COVID-19 pandemic, traditional, face-to-face learning and clinical research were converted into online formats. An hybrid environment of traditional methods and novel technological tools has emerged in readiness for future pandemics that allows for virtual learning with concurrent recognition of the need to provide for interpersonal interactions.

## 1. Introduction

As circumstances change and the COVID-19 (Coronavirus SARS-CoV2 disease) restrictions come and go, it is important for educators to examine how they deliver their message to their students [1,2,3]. There has emerged a need to be somewhat proactive in planning the delivery of education and educational material that respects the inherent risks for both students and teachers, at all levels, from contracting the infection [4,5]. Early in the pandemic, it was recognised that “…academics have a unique opportunity, and a moral duty, to immediately start conducting in-depth studies of current events*…*” and respond to these circumstances [6].

The COVID-19 pandemic was so named by the World Health Organisation (WHO) in March 2020 [7], and since then, the world has experienced the Alpha variant, the Delta variant, which was less transmissible than the Omicron variant but was more virulent; and subsequently, the current Omicron variant [8] which is more transmissible but less virulent, possibly due to the widespread uptake of vaccination against COVID-19 [9]. Some argue that the pandemic, with the Omicron variant, is changing to an endemic situation that will be long-lasting [9].

In Australia, as in other regions, the numbers of infected individuals and those in hospital and intensive care are declining [10], which appears to add credence to the concept that the peak has passed. It is far too early to become complacent as one cannot predict what morphological change to the virus may occur [11] and what impact such changes may have. Vaccination, including quadruple or even quintuple inoculation, including booster “shots”, as advocated for immunocompromised individuals, cannot adequately predict the expected clinical course, despite the improved prognosis [12]. What is apparent is that education and educational techniques have had to accommodate the virus within the community and have had to adjust to public health considerations [13,14,15,16] to deliver optimal educational quality within the current situation. There also has emerged a need to adapt research methods to accommodate the new reality that the pandemic has impacted such practices due to the direct influences of the infection, both on staff and patients, and the delivery of healthcare in its broader consideration, including the process of diagnostics in which telehealth has impacted and continues to impact.

The use of telemedicine and telehealth preceded the pandemic, and in 2019, prior to the first case of COVID-19 being declared, telehealth was already receiving widespread consideration beyond simply being an answer for health delivery to those living in remote areas or those deprived of access to traditional medical care [17]. The application of telehealth included the use of applications for widely used devices to track a person’s exercise activities, the number of steps taken and other physiological parameters [17,18]. With the declaration of the pandemic, the necessity to develop further the technology, encompassed in telemedicine, became an absolute priority [19,20]. Telehealth transcended the delivery of healthcare [21], acknowledging that its use in this domain was greatly enhanced [22], and the application of this technology was further developed beyond that which predated the pandemic to become integral to more than just healthcare and was adapted to medical education, the teaching of students and trainee doctors, virtual diagnosing and the conduct of research methods and became the widely accepted norm, to protect all those within the medical profession, from students through to senior consultants [23,24,25].

The aim of this qualitative study was to evaluate the consequences of COVID-19 in the context of the authors’ experiences in research, education, clinical management and diagnosis with a review of the literature relevant to the authors’ fields of expertise spanning multiple institutions.

## 2. Materials and Methods

A qualitative experiential review was conducted of the impact of COVID-19 on health research, education and diagnosis with a particular focus on stroke research, diagnosis and intervention and the effects of COVID-19 on health sciences education, teaching and diagnostics. The material contained within this paper reflects true experience, which was heavily underwritten by the extensive literature cited in support of that being presented.

The results and the discussion will be presented in three sections: (i) the impact of COVID-19 on clinical research and trials with a focus on stroke; (ii) the impact of COVID-19 on academic teaching with a focus on health sciences; and (iii) the impact of COVID-19 on diagnostics.

## 3. Results and Discussion

### 3.1. The Impact of the COVID-19 Pandemic on Clinical Research and Trials

The aim of clinical research has been to “*yield fruitful results for the good of society*” [26] by improving the health of the population [27] through the conducting of quality experiments on human subjects who have given their voluntary consent to be so involved. The COVID-19 pandemic has challenged the ability to conduct clinical trials and investigator-led research in accordance with these principles. During the pandemic, patient recruitment into clinical trials declined due to: social distancing rules; clinic closures designed to reduce the spread of SARS-CoV-2 infection; patient fear and reluctance to attend face-to-face visits; research staffing shortages; and diversion of research staff to participate in COVID-related activities. Older patients and vulnerable groups, such as culturally and linguistically diverse populations, who are historically under-represented in clinical trials, were particularly affected [28,29]. All of these factors have negatively impacted the sustainability and the viability of being able to conduct clinical trials, including stroke clinical trials as undertaken at various medical institutions, where the commencement of new clinical trials has been delayed and/or patient enrolment into existing trials was suspended [30], in order to ensure patient and staff safety. A need to adapt rapidly to virtual patient and site monitoring visits, new modes of delivery, investigational product and remote data collection became critical for ongoing patient recruitment to satisfy target number expectations and for the successful completion of existing clinical trials [30,31].

Prior to 2019, recruitment into stroke clinical trials at the authors’ institutions was conducted in a traditional face-to-face encounter in which the study protocols were simultaneously explained to patients and families, provided with relevant reading material and given the opportunity to ask, and have answered, questions prior to voluntarily agreeing to participate within the protocol and signing an ethics-committee-approved written informed consent document. Of the twelve acute/post-acute stroke trials being conducted at Liverpool Hospital, within the South Western Sydney Local Health District, at the start of 2019, the majority were institutionally developed projects rather than pharmaceutical industry-sponsored studies. Five were hyper-acute stroke trials involving different intravenous thrombolysis options or other novel hyper-acute therapies in which the study enrolment was time-critical. Due to the time requirements of several of these trials, the decision-making processes had already been simplified, pre-pandemic, with the New South Wales Guardianship Tribunal and with the local ethics committee approval, to obtain verbal consent, for example, by telephone, from a person responsible, followed by a delayed, subsequent confirmation and completion of the written consent that required a signature to be provided within the following 24–48 h. The peak of the COVID-19 pandemic dramatically changed the consent process, as during this time, research staff were asked, and/or had requested, to work from home, and family members, including next of kin, were refused hospital visitation access, making the traditional informed consent processes, such as signing a study continuation written consent form, extremely difficult. This was even more challenging for post-acute trials that did not have an initial verbal consent process in place. The stroke clinical trials unit at Liverpool Hospital experienced the premature closure of recruitment into one post-acute trial, sponsor withdrawal from another and the temporary suspension of recruitment into two hyper-acute and three post-acute stroke trials. Although recruitment into the remaining hyper-acute stroke trials was technically unaffected, the number of acute stroke presentations to the Emergency Department at the hospital, within the trial treatment windows, significantly declined [32]. This reduction in patient numbers across all stroke clinical trials was potentially catastrophic from the financial point of view of the committed research team. Unexpectedly, the pandemic also saw a dramatic rise in novel therapies to treat COVID-19-related illnesses. These studies urgently required experienced research staff and allowed the stroke trials unit to remain financially viable through the part-time and/or full-time deployment of the majority of the research unit staff to perform COVID-19 research, a positive sign of a mutually beneficial collaboration within the local health district, at a time of crisis. The COVID-19 pandemic also saw three of the stroke clinical trials staff, from the Liverpool Hospital, resign to commence work in other departments or external institutions.

The stroke clinical trials unit and the local health district ethics and research department had to adapt rapidly to the COVID-19 pandemic with the implementation of new methods of running trials, such as electronic informed consent forms, remote access to patient hospital records, for data collection by research staff, study protocol amendments that enable virtual patient visits, home pathology blood collection and study drug delivery and the acceptance of remote site monitoring visits by trial sponsors and clinical research organisations. These modifications enabled existing hyper-acute stroke trials to continue recruitment and patient follow-up to their completion. They converted post-acute stroke clinical trials into virtual telehealth platforms, the prime example of which was the Modafinil in Debilitating Fatigue After Stroke 2 (MIDAS2) trial. There were significant delays in the recommencement of MIDAS2, due to COVID-19-related issues in the study drug preparation and delivery, with existing supplies of the study medication having passed their expiry dates. MIDAS2 was the clinical stroke unit’s first fully virtual, post-acute stroke trial, with no requirement for any face-to-face patient visits.

The modification of a clinical trial, such as MIDAS2, from face-to-face engagement to fully virtual due to the consequences of the COVID-19 pandemic is a direction that could be applied to existing clinical studies and considered for future trials. Innovations including electronic completion of consent; drug therapy delivery and/or administration and/or pathology sample collection at a patient’s usual place of residence; and online electronic clinical assessment and data collection are convenient for research staff and patients, an open opportunity for patients living in rural or remote regions to participate in clinical research and allow for trial sites to remain active in the setting of a future pandemic.

### 3.2. The Impact of COVID-19 on Academic Teaching

In 2022, many countries started easing the governmental measures that had been taken to counteract the risk of disease spreading, with almost no travel restrictions nor mandatory quarantines for travellers being enforced.

In Australia, there are no longer bans on social distancing and public gatherings; schools and universities restarted face-to-face interactions with students; businesses have re-opened, and most organisations stopped asking employees to work from home, as was previously the case in the COVID-19 pandemic [33,34]. In the health sciences, it remains mandatory for students and staff, attending Australian hospital facilities, both public and private, to wear a face mask whilst at work.

During the COVID-19 pandemic, authorities in many countries mandated lockdowns or curfews to curtail the spread of the infection [35], resulting in a negative, worldwide effect on businesses, education, health and tourism [36,37]. In preparation for future outbreaks, there has been a greater emphasis on improving online education in medical, allied health and biomedical sciences. The COVID-19 pandemic saw the rapid adaptation and the integration of newer models of learning and teaching methods, such as competency-based medical education (CBME) and augmented reality (AR), into health education. Despite concerns that substituting traditional face-to-face teaching with online learning would have a negative impact on student well-being, the process has resulted in education programs that are potentially more interactive, show medical procedures in real situations, provide concise information and provide three-dimensional virtual tools to mimic live situations [38].

#### 3.2.1. New Models of Learning, Teaching Methods

##### Competency-Based Medical Education

Competency-based medical education (CBME) is an outcomes-based approach to the design, implementation and evaluation of educational programs and the assessment of students based on measurable competencies or observable abilities [39]. It differs from traditional training in its focus on an individual (or individuals) gaining competency for unsupervised practice that is independent of their time in training [40]., A student may be required to perform a minimum number of diagnostic procedures successfully, as opposed to completing a term or rotation in a defined time period. Although CBME predates the existence of COVID-19, it has assumed far greater relevance both during and following the COVID-19 pandemic. During the pandemic, CBME was disrupted by the shutdown of medical programs, the reduction in elective procedures and the redeployment of medical, nursing and allied health staff [41]. The achievement of competency requires the adaptation and/or the development of alternative methods, such as virtual simulation and telehealth [41]. Post-pandemic, CBME allowed for the resumption of direct patient care and real-world experience, although virtual reality has been proven to be effective in a number of domains of education and training, such as competency in basic life support, suggesting that a hybrid model may be the most effective future direction [42].

##### Developing Medical Education with “Augmented Reality”

Augmented reality (AR) has evolved as a preferred modality in medical education because it facilitates a better understanding of how “real world” circumstances and procedures affect medical care. It assumes that, by becoming an active participant in one’s own learning, a student will be more interested and engaged in a subject. The advantages of AR include an opportunity for students to collaborate with one another and actively participate in real-life scenarios. AR has enabled the imparting of the teaching of morphology, surgical methods and the understanding of anatomical substrates, both for teachers and for students. The COVID-19 pandemic saw the successful application of computer and hand-held devices, including tablets and mobile phones, that allowed the continuation of online and remote AR learning, despite academic institutions being shut down. Several AR applications (Apps), including three-dimensional virtual reality-based tools, have been developed during and following the COVID-19 pandemic to enhance appreciation within medical studies [43]. Further research into the utilisation and the educational benefits of existing and future virtual reality-based CBME and AR Apps is warranted.

##### Online Versus Face-to-Face Education

The COVID-19 pandemic triggered the introduction of new methods of both teaching and learning in medical education, with academic institutions accelerating the development of the online learning environment. One such new model is the “flipped classroom”, comprising a blended type of learning mode that incorporates an asynchronous component that encourages medical students to adopt more schedule flexibility together with a synchronous component that offers productive interactions between medical students and faculty members [44,45]. E-learning promoted medical students to better adapt to a web-based medical world that has become more reliant on digital health services, acknowledging that distant learning can potentially hamper personal contact and interaction between medical students and faculty members [46]. The adoption of online learning was a key strategy for ensuring continuity in medical education during the COVID-19 pandemic, though some critics may argue, “Who would want to be treated by doctors who completed their final years of medical education during the pandemic”.

Another problem encountered during the COVID-19 pandemic was the optimal method to examine students while adhering to the imposed restrictions. In many countries, classical, clinical and written examinations were either postponed, cancelled or replaced by online examinations or alternatively by new methods of assessment [47]. Open-book examinations (OBEs) and closed-book examinations (CBEs) options were discussed [48], but due to the inability to organise face-to-face exams, OBEs were considered to be the best option [49]. Medical students were presented with simulated patients and scenarios and asked to respond to set questions based on the provided history and findings from the clinical examination [50]. Modifications to the traditional examinations, implemented due to the pandemic, offered an important opportunity to evaluate alternative modes of medical education and assessment.

##### The Psychological Wellbeing of Students

Even under normal circumstances, medical students suffer from a variety of psychological consequences, such as anxiety, depression and stress, due to a high workload and numerous evaluations and assessments [51]. During the COVID-19 pandemic, levels of anxiety and panic escalated significantly [52,53], demanding that future plans accommodate this most important aspect of medical education.

#### 3.2.2. The Impact of the Post-Pandemic Reality and the Future Direction of Health Sciences

AR and online teaching continue to be consistently utilised post-pandemic. Virtual workshops and online interactive team-based learning constitute a novel form of collaborative post-COVID-19 education. There has been a return to face-to-face teaching, and it will be a matter of time to determine if the lessons learnt from the pandemic, such as social distancing rules, will be incorporated into future teaching models. At the University of Technology (UTS), Sydney, there is a vast laboratory space called the “HIVE”, where at least six to seven courses can be conducted simultaneously. Students use temporal bone headphones and computer tablets and work closely on large machines, concurrently listening to lectures while wearing laboratory coats, masks, eye protection glasses and gloves. There is a fear of overcrowding to accommodate the increased numbers, with a lack of interpersonal communication, as practised in the virtual reality that evolved with the COVID-19 pandemic. This appears likely to persist as academia is emerging from its influences and restrictions.

At the Australian Catholic University, Sydney, where nurses and other allied health staff are being taught, the situation mirrors that at UTS, with students sitting in close proximity to each other, very few, perhaps 1%, opting to wear face masks and a general disregard for those infection control impositions that accompanied the COVID-19 pandemic.

The potential negative impact of overcrowding, either online or face-to-face, on the learning process and imparting of knowledge remains to be determined. The pandemic lasted so long that those responsible for ongoing education may have adopted a “laissez faire” attitude to disease control in the immediate post-pandemic era. There remains the concern that a form of the pandemic still exists endemically and the risk of infection continues but, at government and institutional levels, this appears to be disappearing and the future reaction, to a new variant or strain, is unclear. It also remains to be seen what those responsible for medical education have learnt from the experience of the pandemic or if they have just accommodated its reality without a long-term legacy.

### 3.3. The Impact of COVID-19 on Diagnostics

The COVID-19 pandemic also had a significant impact on diagnostic testing. During the various lockdowns, particularly following the first lockdown, there was a significant reduction in pathology collections, with almost all types of specimens being affected, including biopsy and cytological testing [54,55], with an associated increase in malignancy rates being encountered [56]. Although a return to normality was seen in the post-COVID-19 period, globally, regions which experienced full lockdowns also experienced significant delays in planned cancer surgery and longer pre-operative delays [57]. The disengagement of health professionals, with patients and the community, due to the COVID-19 pandemic, together with the enhanced use of telemedicine, in which the healthcare professional may be totally reliant on patient responses, devoid of the innate element of direct patient contact and interpersonal interaction, has led to genuine concerns of delayed, or missed, diagnoses and inappropriate treatment [58] with the prediction of a significant increase in the number of avoidable cancer deaths [59] prompting a call for urgent health policy interventions [59].

Diagnostic laboratories were also faced with the challenge of how to safely handle blood and tissue specimens that they were asked to process for analysis. Despite the possibility of viral inactivation from the fixatives used, for histopathological specimens, there was no guarantee that the handling of the specimens received from patients who were of unknown COVID-19 infectivity status did not pose a risk. Previous routine specimens were suddenly dealt with as being potentially biohazardous. The use of personal protective equipment (PPE) and the execution and implementation of new safety measures became important to protect courier and laboratory staff from the risk of exposure [60].

Diagnostics were also affected in relation to the availability of diagnostic testing, both clinically and research-based, for non-cancer-related disorders, including diagnostic imaging and intervention studies. In Australia, the first wave of the COVID-19 pandemic was associated with a significant reduction in total imaging services that were available, with general radiology, ultrasound and magnetic resonance imaging (MRI) services being more affected than nuclear medicine and computer tomography (CT) [61]. Imaging facilities, including those with which the co-authors were affiliated, had to adapt to new infection control procedures, including cleaning procedures and PPE use which were based on state government health advice [62]. Intervention services for conditions, such as stroke, were particularly affected in terms of having to implement a number of changes, including pre-procedural COVID-19 testing, restricted access to angiography suites, including: the minimisation of staff numbers; the preparation of suites for COVID-positive patients, including the use of additional shielding equipment; new sanitisation and PPE policies and procedures; and new intubation/extubation procedures for patients not undergoing conscious sedation, such as the use of negative pressure rooms during extubation, where applicable, amongst other major changes [63]. Communication with families who were required to accept restricted visitation rights and delays in the time to the commencement of endovascular thrombectomy for acute ischemic stroke due to large vessel occlusion were additional issues encountered during the peak of the pandemic [63]. Interhospital transfer delays and prolonged door-to-procedure times for COVID-19-positive patients, the latter often due to the performance of CT chest scans, were other consequences of the pandemic waves experienced at the authors’ and other institutions [64]. The interpretation of clinical trial results was also affected, particularly if a COVID-19-positive patient was recruited into a stroke trial, due to the negative effect of concomitant COVID-19 infection on stroke severity and survival [65]. The potential psychological impact of the COVID-19 pandemic on diagnostics radiology staff, especially in the pre-vaccination waves, required strong leadership, proactive communication and the implementation of mental health support services, where available [66].

Diagnostics in other related fields, such as cardiology and vascular surgery, were also affected by the COVID-19 pandemic. Echocardiography and vascular surgery sonographers required a triaging of procedures with a more critical review of the appropriateness of requests to ensure adequate protection and to safeguard sonographers and physician imagers from exposure risk [67]. At the authors’ institutions, there was a necessary reduction and/or delay in diagnostic cardiac investigations, such as transthoracic and transesophageal echocardiography, as well as carotid and vertebral ultrasounds, which often did not proceed until patients were no longer producing COVID-19-positive test results. Training in diagnostic procedures at the authors’ and other institutions became challenging, with competency assessments and volumed-based targets adversely affecting medical trainees’ abilities to meet academic college requirements [68]. Modification of practices, such as a designation of specific roles and the provision of allocated time blocks, to ensure a trainee’s achievement of necessary competencies were implemented [68]. The restructuring of clinical services and the adaptation of online and/or AR teaching techniques became important resource tools.

Diagnostic testing of pulmonary function during COVID-19 required a significant change in clinical and laboratory practice to optimise the safety of healthcare workers and patients [69]. Given that pulmonary function tests (PFTs) result in aerosol generation, risk control strategies that were implemented included avoidance of PFTs in COVID-19-positive or suspected patients, deferment of PFTs in close contact and the performance of tests in a room closed to other internal spaces (that is, not shared) [69]. Home-based oxygen saturation measurements and telehealth spirometry also became a reality [69].

In-hospital diagnostic polysomnography (PSG), necessary to diagnose sleep disorders, especially obstructive sleep apnoea, which is very common, especially in an affluent, largely overweight society, was largely suspended during the pandemic due to the risk of contaminating the diagnostic equipment as well as the risk of transmitting the infection via the reuse of such equipment, despite rigid sterilisation procedures [70,71]. As COVID-19 is a respiratory illness,, the application of PSG and the subsequent application of continuous positive air pressure (CPAP) became problematic, thereby limiting the capacity to diagnose non-compliance or achieve the requisite recording of compliance statistics to determine the efficacy of an intervention or to identify ongoing problems [70].

The COVID-19 pandemic also saw collaborative partnerships between academia and industry, the redeployment of staff from multiple subspecialty disciplines for the common goal of developing rapid diagnostic tests for COVID-19 infection and the establishment of large-scale national vaccination programs. It remains to be seen whether the medical technology and the diagnostic collaborations that occurred, as a result of the global crisis of COVID-19, were temporary or if the accelerated innovations that emerged during the pandemic have established long-lasting global partnerships.

## 4. Conclusions

As a by-product of the COVID-19 pandemic, there has emerged an urgency for a rapid transition in the mode of imparting medical education, the conduct of clinical research and the utilisation of diagnostic services to avoid the constraints that were imposed as a result of mandated, protective intervention.

There is a need to convert many facets of traditional, face-to-face learning into online formats by creating short-term learning opportunities as a substitute for human-contact clinical learning. Effective plans, such as a hybrid environment of traditional methods and adopting novel, technological and virtual tools, should be developed to anticipate the impact of any future pandemics. There is a necessity to implement a hybrid approach to medical education, including diagnostics training and research, that both allows for virtual learning while concurrently recognising the need to provide for interpersonal interactions and an appreciation of the ongoing risks posed by the continued presence of the virus and its variants.

## Data Availability

This study is a review and no new data were created.

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
