# Peer review of "Health Research and Education during and after the COVID-19 Pandemic: An Australian Clinician and Researcher Perspective"

_diagnostics, 2023, doi:10.3390/diagnostics13020289_

Round 1
Reviewer 1 Report
The article is written in not proper manner. It’s lack of creation a hypothesis, methodology part, discussion part, comparison with other research results. More is lack of the theoretical and practical implications of the research results, the research limits further research directions.
Author Response
Thank you for your advice, please see the attachment

Reviewer 2 Report
Dear authors,
The submitted manuscript is well-written and its topic (the impact of the COVID pandemic on health education and research- the Australian perspective) is quite interesting.
I do not have anyhting to add or propose.
Author Response

(The authors gave the same response as above.)

Reviewer 3 Report
I would like to thank the editor for giving me the opportunity to review the manuscript. I commend the authors for the review ‘Health research and education during and after the COVID pandemic…’. The manuscript is well written. I have a couple of comments which I understand will benefit the general reader and improve the quality of the manuscript.
1. Page 4, paragraph 5, first line: Pls change ‘Competency Medical Based Education (CBME)’ to ‘Competency-Based…’.
2. For the benefit of the reader, pls elaborate in brief on ‘Competency-based Medical Based Education’ and ‘Augmented Reality’, and how they were implemented during the COVID-19 pandemic.
3. Page 9: Pls delete the DOI of reference number 35.
Author Response

(The authors gave the same response as above.)

Round 2
Reviewer 1 Report
I did review and decided to reject an article on 7'th December. I can't change my decision.